# Annealing-Dependent Breakdown Voltage and Capacitance of Gallium Oxide-Based Gallium Nitride MOSOM Varactors

**DOI:** 10.3390/ma13214956

**Published:** 2020-11-04

**Authors:** Yu-Li Hsieh, Liann-Be Chang, Ming-Jer Jeng, Chung-Yi Li, Chien-Fu Shih, Hung-Tsung Wang, Zi-Xin Ding, Chia-Ning Chang, Hao-Zong Lo, Yuan-Po Chiang

**Affiliations:** 1Department of Electronics Engineering, Chang Gung University, Guishan, Taoyuan 333, Taiwan; D0527103@cgu.edu.tw (Y.-L.H.); mjjeng@mail.cgu.edu.tw (M.-J.J.); chungyili@mail.cgu.edu.tw (C.-Y.L.); D9828201@cgu.edu.tw (C.-F.S.); M0827102@cgu.edu.tw (Z.-X.D.); M0827103@cgu.edu.tw (H.-Z.L.); 2Department of Electrical and Electronic Engineering, Chung Cheng Institute of Technology, National Defense University, Daxi, Taoyuan 335, Taiwan; 3Green Technology Research Center, Chang Gung University, Guishan, Taoyuan 333, Taiwan; 4Department of Materials Engineering, Ming Chi University of Technology, Taishan, New Taipei City 243, Taiwan; 5Department of Otolaryngology-Head and Neck Surgery, Chang Gung Memorial Hospital, Linkou, Taoyuan 333, Taiwan; 6Graduate Institute of Electro-Optical Engineering, Chang Gung University, Guishan, Taoyuan 333, Taiwan; M0724005@cgu.edu.tw (C.-N.C.); M0424004@cgu.edu.tw (Y.-P.C.); 7Center for Reliability Sciences & Technologies, Chang Gung University, Guishan, Taoyuan 333, Taiwan; 8Department of Radiation Oncology, Chang Gung Memorial Hospital, Linkou, Taoyuan 333, Taiwan; 9Technical Service Center, Industrial Technology Research Institute, Chutung, Hsinchu 310, Taiwan; itri458205@itri.org.tw

**Keywords:** MSM, varactor, Ga_2_O_3_, MOSOM, heterojunction, furnace annealing, breakdown voltage

## Abstract

Our laboratory has previously revealed the use of metal-semiconductor-metal (MSM) varactors against malicious pulses, as well as completed the related verification and measurements of such a circuit. To improve the reliability of this protection module further, in this study, we deposited a gallium oxide (Ga_2_O_3_) thin film in between the Schottky contact electrode to manufacture a metal-oxide-semiconductor-oxide-metal (MOSOM) varactor. However, the thin-film quality and heterojunction interfaces will affect these fabricated varactors in various ways, such as the asymmetry threshold voltage to the variable capacitance characteristics. This study aims to address the issues associated with the inserted oxide thin film, as well as to determine how improvements could be obtained by using an oxygen furnace annealing process. As a result, the breakdown voltage of the MOSOM varactor was further promoted and a more robust anti-surge module was thus realized.

## 1. Introduction

The trend of miniaturization of electronic components is ongoing. This will increase the seriousness of electrostatic discharges (ESDs) or malicious electromagnetic pulse (MEMP) attacks [1,2,3,4,5,6,7]. Furthermore, in modern warfare, electromagnetic interference and electromagnetic bombs can be used to inject high quantities of energy and for specific frequency band attacks [8]. Such attacks will widely damage almost any integrated silicon-related electronic equipment. The use of gallium nitride (GaN), which possesses good material characteristics [9,10], such as a wide energy gap, high electron mobility, and high breakdown voltage, has been previously verified by many laboratories, including ours. Measurements of an anti-surge module using an AlGaN/GaN-based two-dimensional electron gas (2DEG) MSM varactor as the key component have been carried out [11,12]. Some of these anti-surge modules can suppress the attack of malicious electromagnetic pulses to a tolerable level and meet the requirement of MIL-STD-188-125-2 [13]. However, in contrast to our previously published measurement [11], the breakdown voltage of the MSM varactor is about 300 V. In the face of higher voltage and more powerful pulse-current injection, there remain doubts about the property and survival rate. To further improve the breakdown voltage of these key varactor components and to realize a more robust and reliable anti-surge module, an electron-beam evaporated Ga_2_O_3_ thin film was utilized in the deposition process of this article. Thanks to the previous progress of research in MSM varactors, we are able to determine how the design of the electrode structure will affect the breakdown voltage and variable capacitance characteristics [12]. Therefore, we can attempt to further promote the breakdown voltage of an MSM varactor based on an optimized design. Thus, a metal-oxide-semiconductor-oxide-metal (MOSOM) varactor was proposed and fabricated in this study.

Gallium oxide (Ga_2_O_3_) exists in five different crystal phase structures under various growth temperatures and pressure environments [14]. The most stable phase, the β-phase, may be obtained under normal room temperature and atmospheric pressure. Compared to GaN, Ga_2_O_3_ possesses a higher energy band gap (4.85 eV) and breakdown field (8 MV/cm). As a new generation candidate for high-frequency and high-power materials, Ga_2_O_3_ has been intensively researched and has obtained a great deal of attention [15,16]. In recent years, as the realization of high-purity and epitaxial-quality gallium oxide wafers has been achieved, in addition to breakthroughs in doping technology, the application of gallium oxide-related gallium nitride devices has been demonstrated as a highly beneficial system. General preparation methods for achieving better epitaxial quality for gallium oxide wafer are the Czochralski process and metalorganic chemical vapor deposition system (MOCVD) [17,18]. These epitaxial wafers may then be applied in the design and development of new electronic components. Conversely, Ga_2_O_3_ thin films deposited via E-gun may result in some quality issues, heterojunction problems, mismatches in lattice constants, and polarization effects. All of these issues will inevitably impact the capacitance variable characteristic of the varactor and cause related issues. As far as the authors’ limited views are concerned, relevant literature related to MSM varactor research is lacking.

However, an E-gun deposition system is in place in almost every semiconductor factory or laboratory and is still a cost-effective process. Furnace annealing processes are widely used to improve thin-film quality and eliminate lattice defect problems [19]. As the key material to the variable capacitance characteristics of MOSOM varactors, the 2DEG layer should not be harmed by the oxygen furnace annealing process. The annealing time and temperature parameters are, therefore, key issues. According to a reference [20], Ga_2_O_3_ grain growth at dislocations of the nitride epilayers promotes nitrogen removal and oxide formation. Hence, the roughness of the oxidized thin film becomes more serious as the annealing temperature exceeds 900 °C. An inappropriate annealing process may damage the 2DEG layer within the GaN epitaxial wafer and the variable capacitance characteristic of the varactor thus no longer exists. In this study, an oxygen furnace annealing process with designed parameters is applied for improving the thin-film quality of Ga_2_O_3_ and solving the heterojunction problems within an MOSOM varactor. The experimental results show that our annealing process can improve the asymmetry of the variable capacitance, which is rarely studied. In addition, the breakdown voltage of the varactors can be greatly improved relative to our previously published measurement [11]. The proposed MOSOM varactor demonstrates a better capacitance characteristic and a higher breakdown voltage. Therefore, the design of protective varactor modules for future high-frequency front-end circuits used to resist ESD and MEMP attacks will obtain greater attention in future research.

## 2. Experiment

### 2.1. Fabrication of the AlGaN/GaN-Based 2DEG MSM Varactor

The epitaxial wafer used in our experiment for fabricating the MSM varactor was an AlGaN/GaN-based 2DEG heterostructure wafer, which is commonly used to prepare GaN high-power devices. The wafer structure is shown in Figure 1a. To obtain a more detailed manufacturing process of the MSM varactor, we can refer to our previously published literature [12]. Based on previous research results, we know that the design of the electrode structure will affect the maximum capacitance (C_max_), minimum capacitance (C_min_), capacitance conversion ratio (C_ratio_), and breakdown voltage of the varactor. Therefore, we chose to utilize the optimized design determined in previous research results to obtain a unified metal electrode mask pattern. The design of the mask adopts a structure with an electrode length L of 2000 µm and an electrode spacing of 40 µm, as shown in Figure 1b, which was used to prepare the MSM varactor for the following experiments.

### 2.2. Fabrication of the AlGaN/GaN-based 2DEG MOSOM Varactor

Subsequently, an electron-beam evaporation system was used to deposit a Ga_2_O_3_ thin film (with two different thicknesses of 156 and 341 nm) onto the GaN epitaxial wafer. Next, before manufacturing the metal electrode, an oxygen furnace annealing process with different experimental parameters (as shown in Table 1) was applied to complete the different heterojunctions between the Ga_2_O_3_ thin film and GaN wafer. The standard lithography process was then used to deposit the Ni (20 nm)/Au (70 nm) metal electrode to complete the MSM and MOSOM varactors (as shown in Figure 2). After the lift-off process, the final MSM and MOSOM varactors are shown in Figure 3 and the optical microscope images of this component are shown in Figure 4.

### 2.3. Measurement of the Variable Capacitance Characteristics and Breakdown Voltage

Finally, to explore the effects of the heterojunction on the electrical properties of the varactor, the effects of the oxygen furnace annealing process on the improvement of the issue occurring between the heterojunctions, high resistance (using a Keysight B2985A high resistance meter, Keysight Technologies, Santa Rosa, CA, USA), X-ray diffraction (XRD, using a D2 Phaser X-ray diffractometer, Bruker, Billerica, MA, USA), capacitance–voltage (C–V, using an Agilent E4980A LCR-meter, Agilent Technologies, Santa Clara, CA, USA), and current–voltage (I–V, using a Keithley 2410 SourceMeter, Keithley Instruments, Solon, OH, USA) measurements were performed. The increase in the breakdown voltage in the overall system was also investigated.

## 3. Results and Discussion

### 3.1. The Breakdown Behavior of the MSM Varactor after Malicious Pulse Injection

In an anti-surge module study published previously, the MSM varactor was placed in series using a flip chip on a 50-ohm signal transmission line [11]. To test the robustness of this component, we also directly injected a 600-A malicious current pulse into the MSM varactor without a gas discharge tube (GDT) shunted in front of it for reliability testing. Although the MSM varactor could immediately drop down its own capacitance value to cut off the attack energy, the device itself also collapsed (as shown in Figure 5), which demonstrates the necessity of increasing the breakdown voltage of the MSM varactor. Moreover, the GaN epitaxy wafer in this structure cracked on the surface due to the uneven thermal stress caused by the implanted energy and permanent damage occured (as shown in Figure 6). C–V measurements of the breakdown varactor were obtained. It can be seen that the variable capacitance characteristic of this device no longer exists (as shown in Figure 7).

### 3.2. Electrical Properties of the Grown Gallium Oxide

The thicknesses of the gallium oxide thin films deposited using gallium oxide powder with an electron-beam system were approximately 341 and 156 nm (using an Alpha-Stepper, Surfcorder ET3000, Kosaka Laboratory Ltd., Tokyo, Japan), as shown in Figure 8. The XRD measurement results are shown in Figure 9. The peaks can be seen at the 2θ values located at 34.6° (this belongs to the GaN) and 35.12° (this belongs to Al_0.26_Ga_0.74_N) [21,22,23]. After the 500 °C oxygen furnace annealing process, a comparison of the XRD diffraction peaks demonstrates that the full width at half maximum (FWHM) and the peak intensity of the signal did not substantially decrease. This means that the 500 °C oxygen furnace annealing process does not significantly cause damage to the original GaN epitaxial layers underneath the Ga_2_O_3_ thin film. The high resistance measurement results are shown in Figure 10. Comparing the measurement results between the open circuit and the resistance of the Ga_2_O_3_ thin film, the insulation properties of the Ga_2_O_3_ thin film deposited by the electron beam evaporation system can reach more than 50 GΩ. Therefore, it is useful in applying a potential to increase the overall breakdown voltage of the MSM varactor.

### 3.3. Influence of the Ga_2_O_3_ Thin Film on the Variable Capacitance Characteristic of the MOSOM Varactor

The thickness of the Ga_2_O_3_ thin film will significantly affect the capacitance value of the MOSOM varactor under normal signal transmission conditions. Because the MSM varactor is placed in series on the signal transmission path, the capacitance value of the MSM varactor should be as large as possible, such that the signal will be smoothly coupled to the secondary side of the anti-surge module. Therefore, to analyze how the capacitance value may be influenced by the Ga_2_O_3_ thin film, we utilized different thicknesses (156 and 341 nm) for comparison.

In the C–V measurement results, before adding the Ga_2_O_3_ thin film (i.e., retaining the MSM structure), the maximum capacitance value of the MSM could reach about 450 pF (as shown in Figure 11) under normal small-signal transmission conditions (at the millivolt level, 2 MHz). When the 341 nm thick Ga_2_O_3_ thin film was inserted (as in the MOSOM structure), the maximum capacitance value reduced to about 35 pF (as shown in Figure 12). As the thickness of the Ga_2_O_3_ thin film was halved to 156 nm, the maximum capacitance value doubled to approximately 70 pF (as shown in Figure 13).

The experimental results confirm that when the MOSOM varactor is operating under normal working conditions (for small-signal transmissions), all of the biased values are under the threshold voltage, such that neither the anode nor cathode depletion regions can contact the 2DEG layer, in which the equivalent capacitance is extremely small. The overall capacitance of the entire MOSOM varactor was determined by the equivalent forward- and reverse-biased back-to-back Schottky diodes, which are located in series. As the thickness of the Ga_2_O_3_ thin film was halved, the overall capacitance of the MOSOM varactor doubled. This is because all of the equivalent back-to-back Schottky diode capacitances were doubled by halving the oxide layer thickness.

Due to the mismatch of the lattice constants and the influence of polarization effects, the Ga_2_O_3_ thin film deposited by E-gun on the GaN wafer must experience stress and internal electric polarization field at the heterojunction. As a result, the C–V measurements are influenced by this phenomenon, such that the threshold voltage (where the varactor capacitance value suddenly declines) is delayed (as shown in Figure 12 and Figure 13). We can see that the original threshold voltage of the MSM was about ±4.5 V (as shown in Figure 14) and it was symmetrical. However, after the Ga_2_O_3_ thin film was deposited to form the MOSOM structure, the threshold voltage shifted. As the DC bias voltage is sweeping from −15 V to +15 V (as shown in Figure 13) or from +15 V to −15 V (as shown in Figure 15), the critical voltage of the overall capacitance drops from C_max_ to C_min_ are all delayed by about 3 V (as shown in Figure 16). Although this phenomenon of asymmetry does not have a serious impact under small-signal transmission conditions, the reliability and signal distortion of these devices may be more greatly impacted during high-power and high-voltage transmission usage.

### 3.4. Influence of the Oxygen Annealing Process on MOSOM’s C-V Characteristics

Because the 2DEG layer results in the variable capacitance characteristic of the varactor, it should not be damaged after undergoing the oxygen furnace annealing process. Therefore, referring to the ohmic contact manufacturing process of common high-electron-mobility transistor (HEMT) devices, we utilized the lower temperature of 500 °C for the oxygen annealing experiment design. Additionally, a higher temperature of 900 °C and a longer annealing time were also utilized. As the anneal duration time is increased, the color of the gallium oxide thin-film surface tends to fade and uneven, rough particles appear on the surface of the Ga_2_O_3_ thin film. This phenomenon is most obvious in the sample obtained under 900 °C and 30 min of annealing (as shown in Figure 4).

Considering the influence of this process on the capacitance, under 500 °C and 2 min of annealing (as shown in Figure 17), interesting phenomena occur in the C–V pattern. First, the sweeping DC bias voltage must be expanded to ±20 V to fully measure the variable capacitance characteristics. Second, the threshold voltage of the MOSOM varactor obviously returns to a symmetrical structure, which occurs in the range of approximately ±15 V. Third, the changing pattern of the capacitance value occurs in two stages. A slight decrease occurs at about ±5 V, and the capacitance then drops completely at ±15 V. The experimental results show that the stress problem between the heterojunction and the influence of the polarization effect have been improved by the annealing process, such that the threshold voltage is again symmetrical. However, the changed pattern of the capacitance value occurs over two stages and the threshold voltage greatly increases from 4.5 to 15 V. These phenomena should be further investigated in the future. In addition, as the C–V measurement signal frequency is decreased, it can be seen that the Batman-like capacitance characteristics once again appear, i.e., at higher frequencies of 1–2 MHz, the capacitance side-values are suppressed to the overall lower-capacitance value.

After applying the 500 °C oxygen furnace annealing process for 4 min, the issues between the heterojunctions can not only be improved but a symmetrical threshold voltage state is restored (as shown in Figure 18). Furthermore, as the C–V measurement signal frequency decreases, the Batman-like capacitance characteristic no longer appears and the capacitance variation becomes more stable. In addition, the threshold voltage is slightly increased from about ±4.5 to ±5 V compared to the MSM varactor without the Ga_2_O_3_ thin film. These results agree with our expectations. After the Ga_2_O_3_ thin film is inserted, the overall thickness of the wafer structure is correspondingly thicker. Neither the anode nor the cathode depletion regions can contact the 2DEG layer under a ±4.5 V DC bias voltage condition. To ensure that the depletion zone under the reverse-biased electrode may expand into the 2DEG layer and cause a sudden drop in the overall capacitance, the bias voltage must be slightly increased for a thicker device. The influence of the different annealing parameters on the C–V characteristics of the MOSOM devices are summarized in Figure 19.

In terms of the MOSOM varactor after undergoing 900 °C and 30 min of annealing, the variable capacitance characteristic no longer exists. This indicates that the 2DEG layer and wafer structure have been damaged by the high-temperature and long-duration oxygen furnace annealing process. According to reference [20], the Ga_2_O_3_ grain growth at dislocations of the nitride epitaxial layer promotes nitrogen removal and oxide formation. The roughness for oxidized GaN/Si wafer becomes more serious as the annealing temperature exceeds 900 °C. Thus, the high-temperature and long-duration oxygen furnace annealing process results in the oxidation effect, resulting in penetration into the shallow 2DEG layer and a rougher Ga_2_O_3_ thin-film surface (Figure 4e). Considering the annealing temperature and time, it is still necessary to optimize the process requirements between the improvement of the heterojunction and the maintenance of the variable capacitance characteristic.

### 3.5. Influence of the Oxygen Annealing Process on the MOSOM’s I-V Characteristic

In the I–V measurement results and breakdown image of the MSM varactor (as shown in Figure 20a,b), the best breakdown voltage of the original MSM varactor is 253 V. Because the MSM varactor is placed in series with the signal transmission path, the secondary side of the anti-surge module will follow the gate of the HEMT or metal-oxide-semiconductor field-effect transistor (MOSFET). Compared with the unprotected transistor or power devices, the MSM varactor breakdown voltage can reach 253 V. A more significant improvement of this characteristic could potentially be obtained.

When the Ga_2_O_3_ thin film (156 nm) is added without annealing, the breakdown voltage of this device can reach 341 V (as shown in Figure 21a,b). After applying 500 °C and 2 min of oxygen furnace annealing, the breakdown voltage of the MOSOM varactor can be increased to 683 V (as shown in Figure 22a,b). After lengthening this process to 4 min, a 421 V breakdown voltage can be obtained (as shown in Figure 23a,b). It can therefore be confirmed from the experimental data that gallium oxide improves the breakdown voltage of these MSM varactors (as shown in Figure 24).

If the thickness of the Ga_2_O_3_ thin film is increased to 341 nm, the breakdown voltage can be increased to 631 V (as shown in Figure 25a,b) without undergoing oxygen furnace annealing to overcome the issues associated with the heterojunction, although the maximum capacitance will decrease to 35 pF. Therefore, depending on the frequency band requirements of the protected circuit, we can adjust the thickness of the gallium oxide thin film to determine the capacitance value required for the varactor to obtain better signal transmission. In addition, the issues associated with the thin-film quality and heterojunction can be overcome via the oxygen furnace annealing process. This enables the varactor to transmit signals in a normal operating mode and can immediately cut off the surge energy when attacked by pulses. With the enhanced ability of the breakdown voltage, the varactor can resist the extremely quick rising time attacks because the GDT shunted in front of the varactor will have enough time to start and will become the main pathway to release the energy surge. Therefore, the robustness of the overall anti-surge module could be enhanced using this strategy.

## 4. Conclusions

At first, experimental results show that the thin-film Ga_2_O_3_ deposited via E-gun causes a C–V threshold voltage delay and asymmetry in the fabricated MOSOM varactors. After, application of the appropriately designed oxygen furnace annealing process can eliminate the problem of the asymmetry of the MOSOM threshold voltage, improve the influence of polarization effects, and reduce the stress between the heterojunctions. Most importantly, the breakdown voltage of the MOSOM varactor can be significantly improved up to 683 V, which is beneficial against the introduced MEMP attacks.

The maximum capacitance value of the original MSM varactor was 450 pF and the breakdown voltage was only 253 V. After adding a 156 nm thick gallium oxide film without annealing, the maximum capacitance value of MOSOM reduced to 70 pF (which is good enough for the GPS signal) and the breakdown voltage increased to 341 V. After applying 2 min of 500 °C oxygen furnace annealing to the MOSOM varactor (with a 156 nm thick Ga_2_O_3_ thin film), the maximum capacitance value remained as 70 pF while the breakdown voltage increased to 683 V. However, as this annealing process was lengthened to 4 min, the maximum capacitance remained at 70 pF while the breakdown voltage decreased again to 421 V. Thus, optimization steps could be made for a post-oxygen-annealing process to fabricate the GaN/Ga_2_O_3_ MOSOM varactors. Quantitative analysis of the strain/stress and polarization effects of the annealed Ga_2_O_3_ thin film could be performed for future perspectives.

## Figures and Tables

**Figure 1 materials-13-04956-f001:**
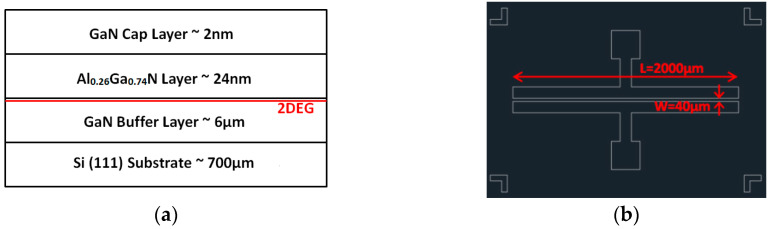
(**a**) Epitaxial structure of the AlGaN/GaN-based two-dimensional electron gas (2DEG) wafer; (**b**) mask design of the optimized metal-semiconductor-metal (MSM) varactor.

**Figure 2 materials-13-04956-f002:**
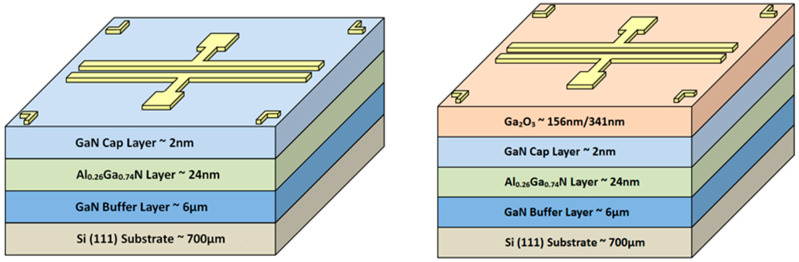
3D structure of the MSM and metal-oxide-semiconductor-oxide-metal (MOSOM) varactors.

**Figure 3 materials-13-04956-f003:**
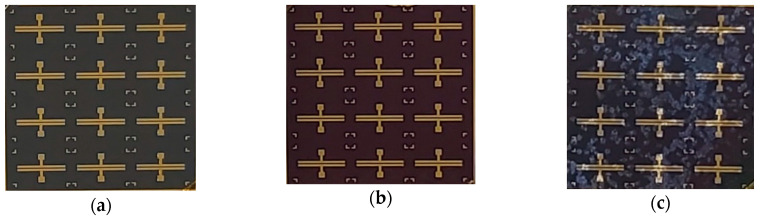
Images of the (**a**) MSM varactor, (**b**) MOSOM varactor (without annealing), and (**c**) MOSOM varactor (with 900 °C annealing).

**Figure 4 materials-13-04956-f004:**
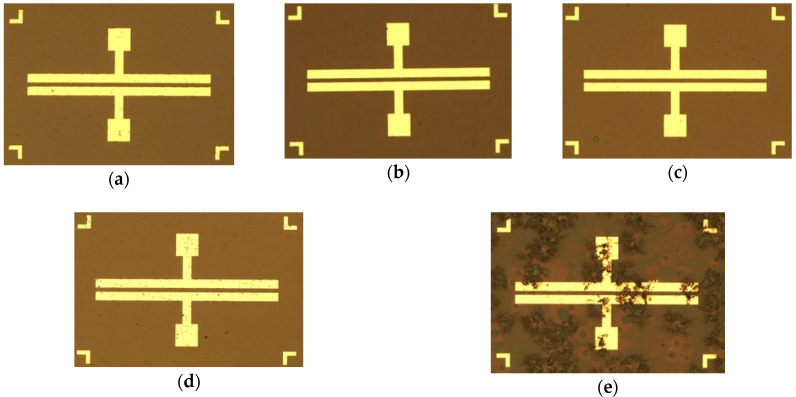
Optical microscope images of the (**a**) MSM varactor, (**b**) MOSOM varactor (without annealing), (**c**) MOSOM varactor (with 500 °C and 2 min of annealing), (**d**) MOSOM varactor (with 500 °C and 4 min of annealing), and (**e**) MOSOM varactor (with 900 °C and 30 min of annealing).

**Figure 5 materials-13-04956-f005:**
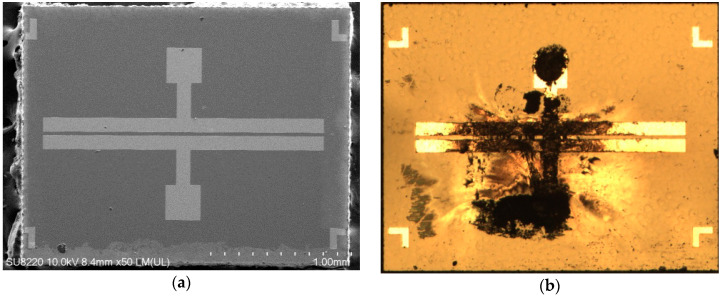
Images of the (**a**) normal MSM varactor and (**b**) breakdown state after the injection of a 600-A malicious current pulse.

**Figure 6 materials-13-04956-f006:**
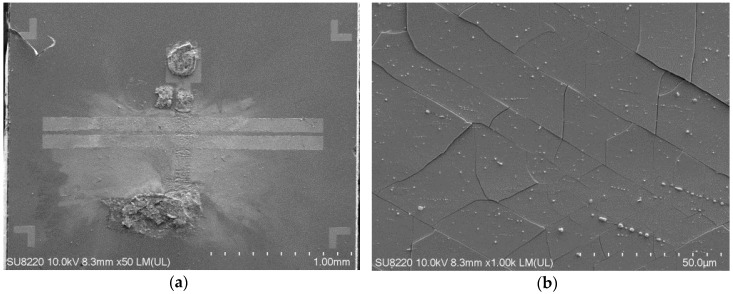
SEM images of the (**a**) breakdown-state MSM varactor and (**b**) GaN epitaxial wafer structure cracked on the surface.

**Figure 7 materials-13-04956-f007:**
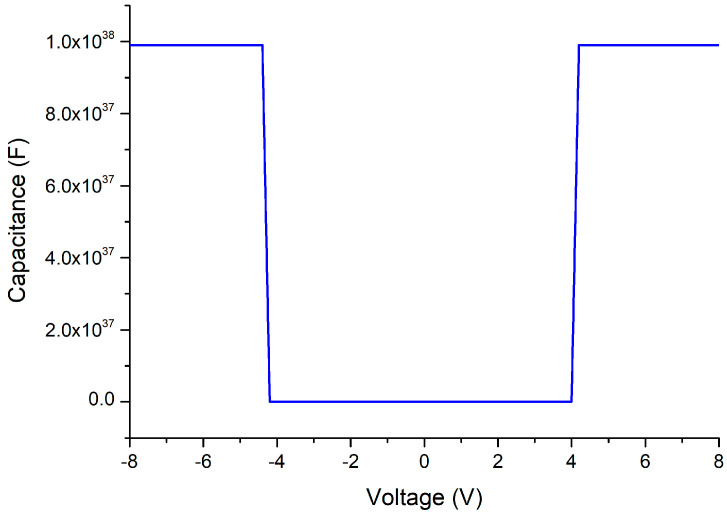
Capacitance–voltage (C–V) measurements of the breakdown MSM varactor.

**Figure 8 materials-13-04956-f008:**
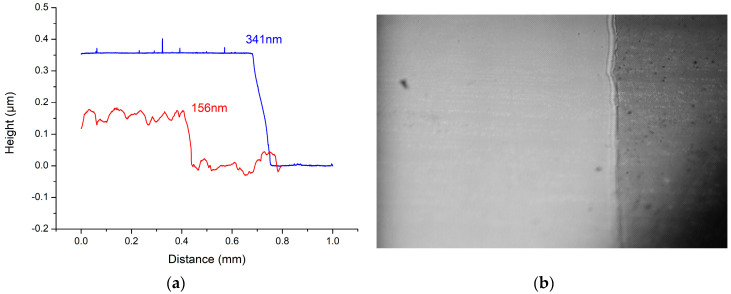
Alpha Step measurements of the Ga_2_O_3_ thin film for (**a**) 341 and 156 nm thickness, and (**b**) an optical microscope image of the thin film surface.

**Figure 9 materials-13-04956-f009:**
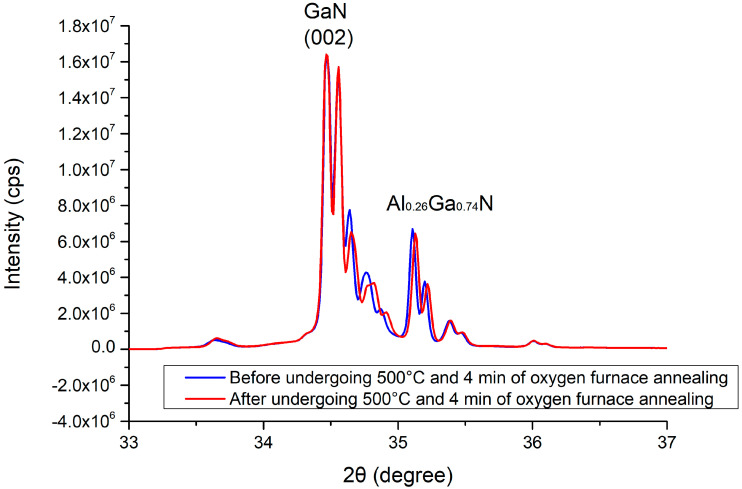
XRD measurements of the Ga_2_O_3_ thin film deposited on the GaN wafer before (blue curve) and after (red curve) undergoing 500 °C annealing for 4 min.

**Figure 10 materials-13-04956-f010:**
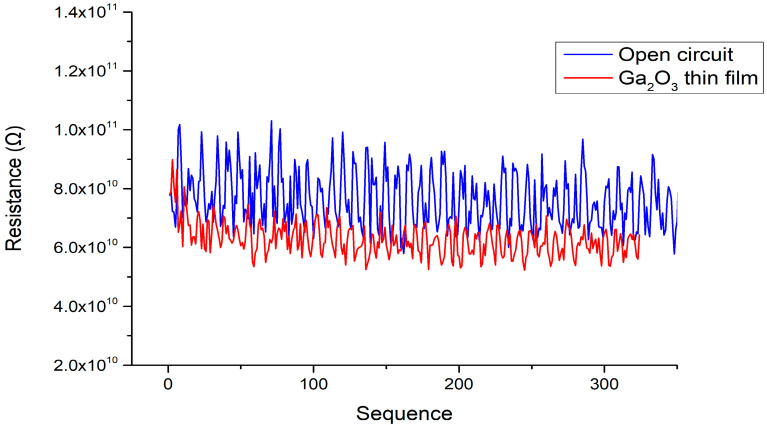
Comparison of the sequence (in continuous loop mode) high-resistance measurement results between the Ga_2_O_3_ thin film and open circuit.

**Figure 11 materials-13-04956-f011:**
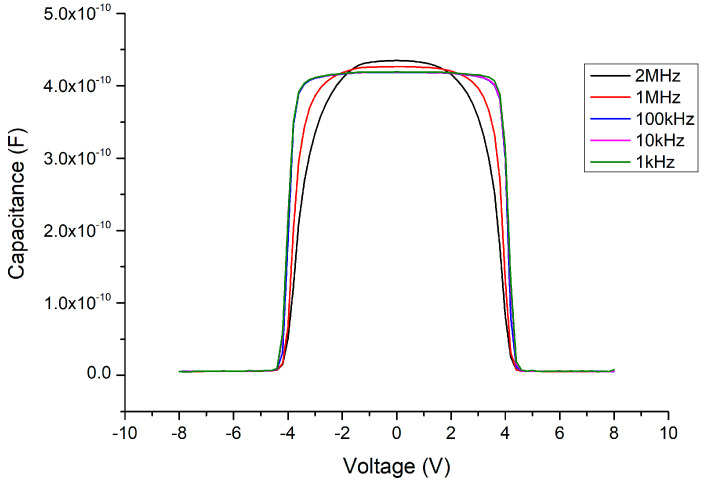
C–V measurement results of the MSM varactor.

**Figure 12 materials-13-04956-f012:**
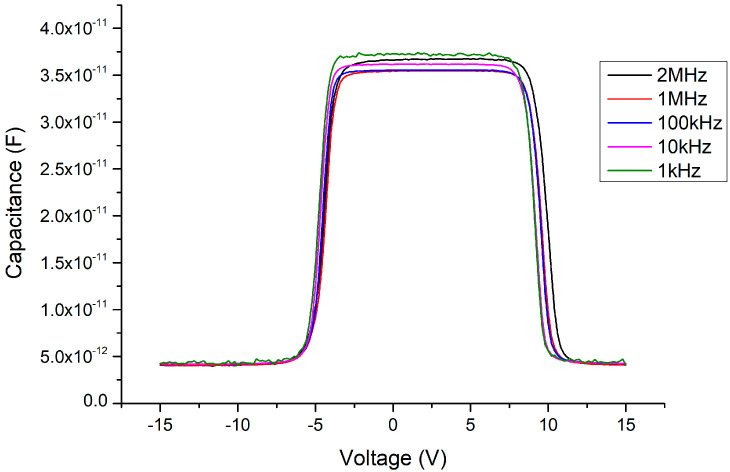
C–V measurement results of the MOSOM varactor (with a 341 nm thick Ga_2_O_3_ thin film) from −15 to +15 V.

**Figure 13 materials-13-04956-f013:**
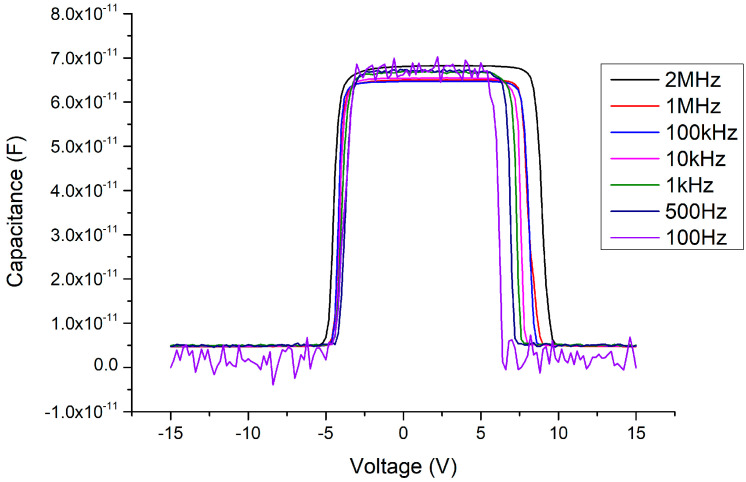
C–V measurement results of the MOSOM varactor (with a 156 nm thick Ga_2_O_3_ thin film) from −15 to +15 V.

**Figure 14 materials-13-04956-f014:**
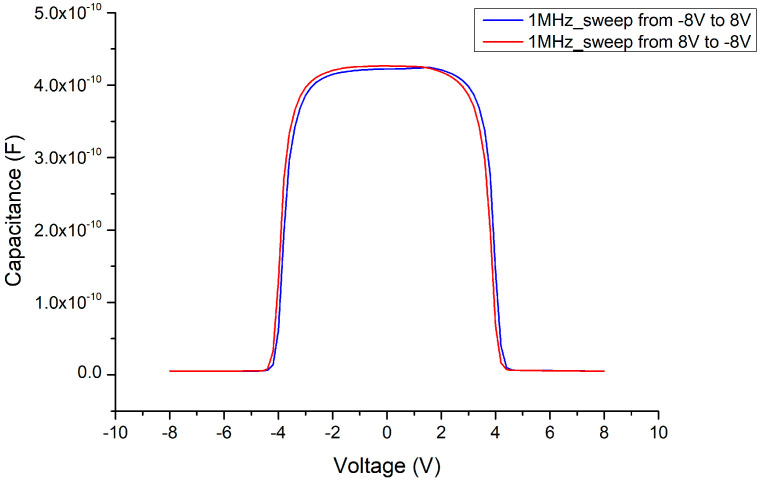
C–V measurement comparison of the MSM varactor between ±8 V.

**Figure 15 materials-13-04956-f015:**
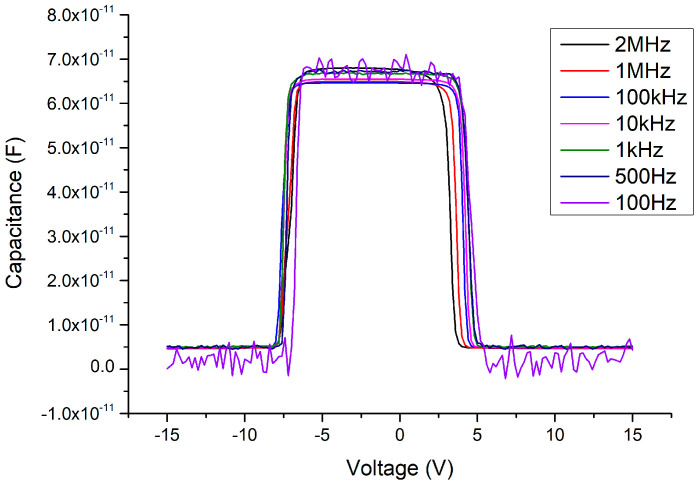
C–V measurement results of the MOSOM varactor (with a 156 nm thick Ga_2_O_3_ thin film) from +15 to −15 V.

**Figure 16 materials-13-04956-f016:**
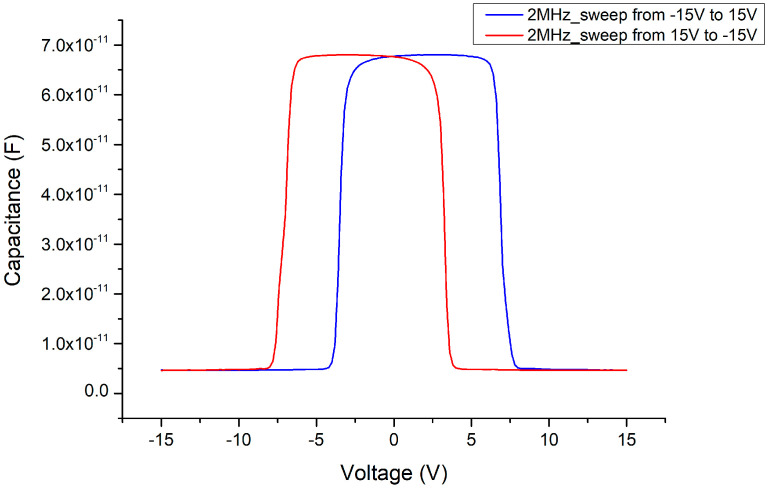
Threshold voltage of the MOSOM varactor.

**Figure 17 materials-13-04956-f017:**
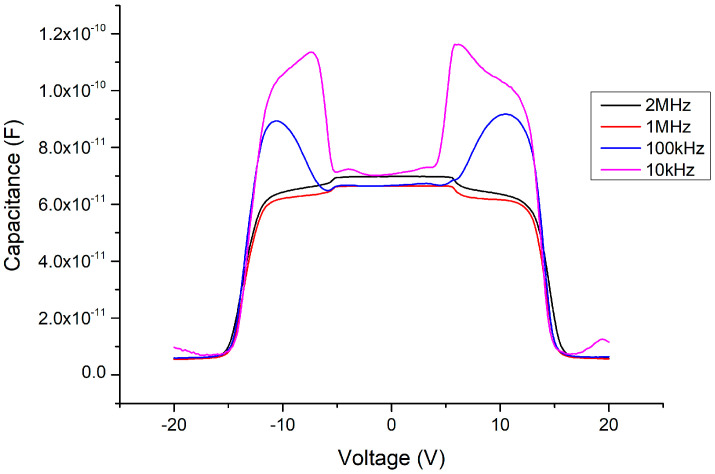
C–V measurement results of the MOSOM (with a 156 nm thick Ga_2_O_3_ thin film) after 500 °C and 2 min of annealing.

**Figure 18 materials-13-04956-f018:**
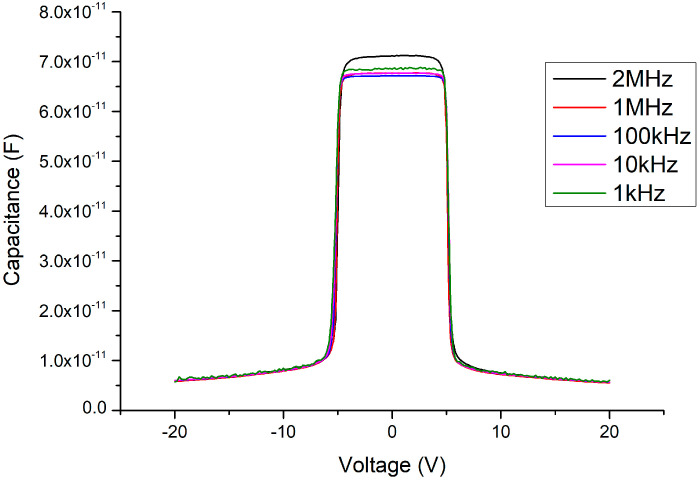
C–V measurement results of the MOSOM (with a 156 nm thick Ga_2_O_3_ thin film) after 500 °C and 4 min of annealing.

**Figure 19 materials-13-04956-f019:**
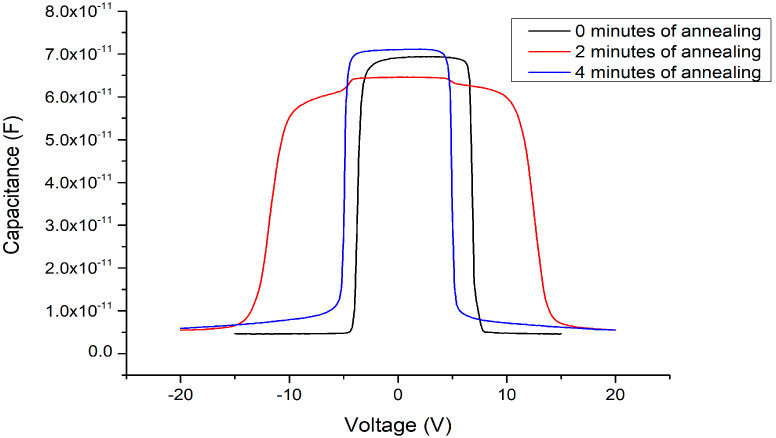
C–V measurement results of the MOSOM obtained using different annealing parameters.

**Figure 20 materials-13-04956-f020:**
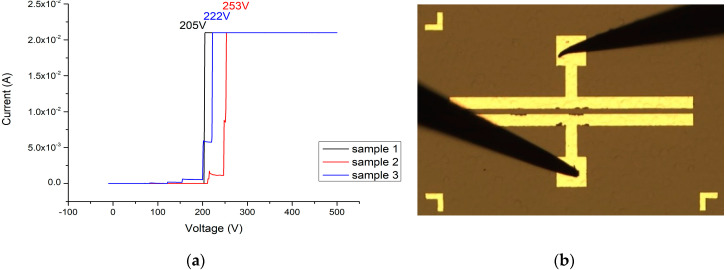
(**a**) I–V measurement results (sample 1 in black, sample 2 in red, and sample 3 in blue) and (**b**) breakdown image of the original MSM varactor.

**Figure 21 materials-13-04956-f021:**
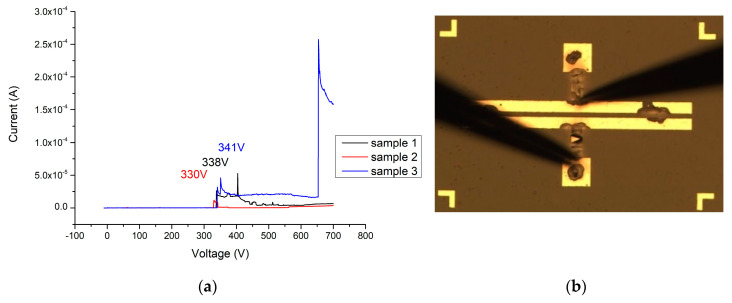
(**a**) I–V measurement results (sample 1 in black, sample 2 in red, and sample 3 in blue) and (**b**) breakdown image of the MOSOM varactor (with a 156 nm thick Ga_2_O_3_ thin film) without the annealing process.

**Figure 22 materials-13-04956-f022:**
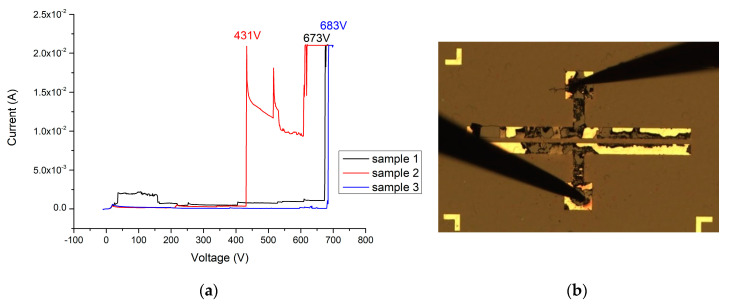
(**a**) I-V measurement results (sample 1 in black, sample 2 in red, and sample 3 in blue) and (**b**) breakdown image of the MOSOM varactor (with a 156 nm thick Ga_2_O_3_ thin film) after the oxygen furnace annealing process at 500 °C for 2 min.

**Figure 23 materials-13-04956-f023:**
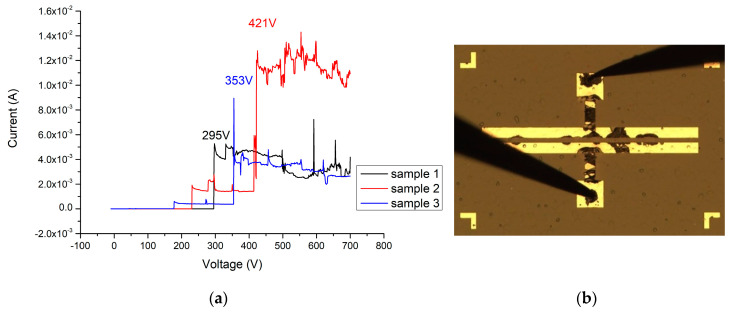
(**a**) I–V measurement results (sample 1 in black, sample 2 in red, and sample 3 in blue) and (**b**) breakdown image of the MOSOM varactor (with a 156 nm thick Ga_2_O_3_ thin film) after the oxygen furnace annealing process at 500 °C for 4 min.

**Figure 24 materials-13-04956-f024:**
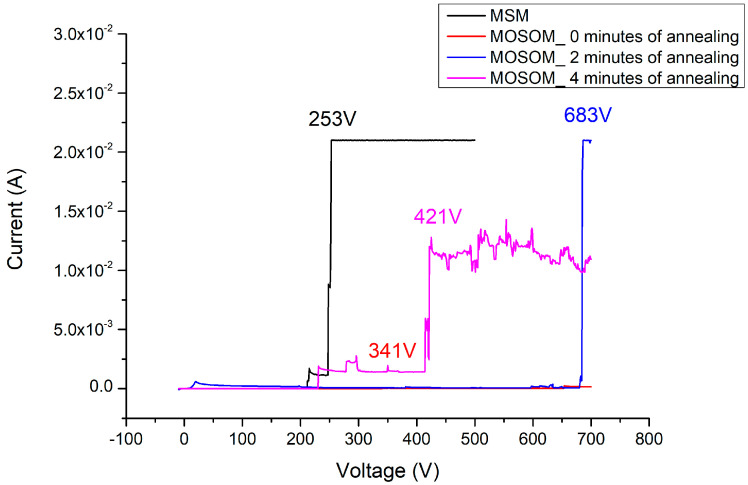
I–V measurement results of the MOSOM varactors under different annealing parameters.

**Figure 25 materials-13-04956-f025:**
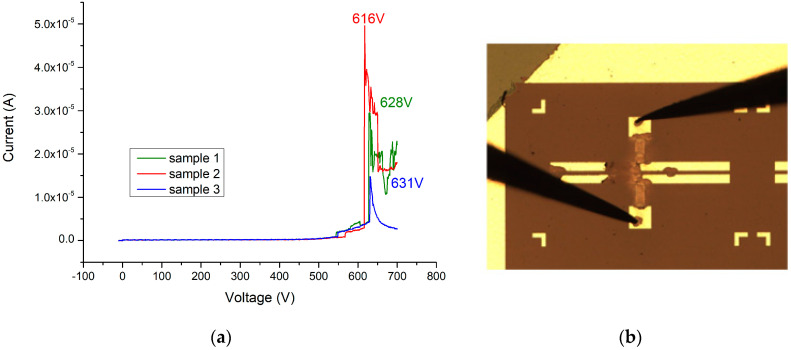
(**a**) I–V measurement results and (**b**) breakdown image of the MOSOM varactor (with a 341 nm thick Ga_2_O_3_ thin film) without annealing.

**Table 1 materials-13-04956-t001:** Experimental parameters used in the oxygen furnace annealing process.

Parameters	Units	1	2	3
Temperature	°C	500	500	900
Time	min	2	4	30

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
