# Peer review of "Annealing-Dependent Breakdown Voltage and Capacitance of Gallium Oxide-Based Gallium Nitride MOSOM Varactors"

_materials, 2020, doi:10.3390/ma13214956_

Round 1
Reviewer 1 Report
The papers presents a study on the electrical characteristics of gallium-based VARACTORS. The study is complete, but the presentation of the information is a bit obscure. Quality of the figures must definitely be improved.
Reviewer 2 Report
The paper deals with the electrical characterization and investigations of the impact of annealing procedure onto I-V, and C-V characteristics of GaN-based varactors. Although the results presented in the manuscript may be found interesting, but the technical presentation of the results is very weak and the text must be thoroughly revised in terms of English.
The following issues need to be addressed by the Authors before the paper can be considered to publication in Materials:
1) The text must be thoroughly revised - even the first sentence in the Abstract includes the mistake, i.e. "...has previously proposed and fabricated the use of metal-semiconductor-metal..."
2) Table 1 need technical improvement; I advise putting units, i.e. (oC) and (min) after the name of the parameter in column one, while in the remaining columns put only numbers. Moreover, the word "duration" does not need the following word "time".
3) What is the "standard lithography" process...?
4) Fig. 7 - the 0Y axis name should be "Capacitance".
5) The labels of Figs. 8, 10, 11, 12, 13, 14, 15, 16, 17, 18, 19, 20, 21 are misleading. Please prepare elegant forms of the labels and add some more explanation in appropriate captions. For example, what is the "thickness1" and "thickness2" (Fig. 8a), what are "test4_2MHz_1" (Fig. 11), "test4_100kHz_3" (Fig. 12), "S11_2MHz+-15 to 15_R8" (Fig. 16), and so on ? Please make corrections in all labels.
6) The quality of Figs. 8, 20, 26 is poor. It seems like some Figs. are pasted in high resolution, and some not. Please, be consistent in the form, dimensions, and quality of all Figures pasted in the text.
7) In order to save some space, I advise putting Figs. 8a and 8b into one Figure. Moreover, Fig. 9 also needs to be corrected - it is hard to distinguish both spectra. It would be necessary to split horizontally the data. Also, the intensity should be substituted by appropriate chemical composition - the difference of the peaks intensity should be clearly visible by the appropriate formation of the Figure itself.
8) Why Author claims when the peak intensity of the signal has increased the improvement in the lattice dislocation is raised (page 6, line 150). Do the Authors provide nay experimental result, prove or any reference to that statement?
9) Names of 0Y axes of the Figs. 22, 23, 24, 25, 26 and 27 are improper. The data do not show the "measure current" but the "current". What are "S1", "S2", and "S3" and other labels? The captions need to be formulated more clearly.
10) Are the Authors sure that the picture of the broken structure needs to be presented each time with the Figure showing I-V characteristic?
Reviewer 3 Report
The manuscript attempts to explain the higher breakdown voltage and variable capacitance characteristics of the Gallium oxide based Gallium Nitride MSM/MOSOM varactors. The research topic is of interest and importance to resist the electrostatic discharges or malicious electromagnetic pulses attacks. The obtained results are very interesting. However, the explanation of the results in terms of physics has not been done sufficiently in the manuscript. It is not clear how this manuscript improves the understanding of MOSOM varactor physics. My specific comments on this manuscript are given below.
- Abstract: The main work was focused on the breakdown voltage and variable capacitance characteristics. The abstract does not reflect any statement on the capacitance characteristic. Therefore, it is difficult to understand the aim/goal of this research work form the abstract.
- Introduction:
- There is no information on the previous results available in the literature. Therefore, it is hard to know what research works have been done on these topics so far.
- Page no 2, line no 52 (‘Thanks to the previous progress of research in MSM varactors, ….capacitance characteristics’) needs a suitable reference.
- Experiment: It is mentioned that ‘To obtain a more detailed … we can refer to our previously published literature.’. Where is the reference no?
- Results and discussion:
- Based on figure 6(b), authors have suggested that the GaN epitaxy wafer structure cracks along the ‘lattice direction’. Did the authors check the cross-sectional view of the cracked sample by SEM/TEM (specimen with cracks should be prepared by Focused ion beam(FIB))? Otherwise, it would be better to use the word ‘cracks on the surface’.
- ‘In terms of the MOSOM varactor after undergoing 900 °C and … no longer exist.’ (Page no13, line no 254). What is the threshold temperature and duration of time for the oxygen furnace annealing process? What was the motivation to use 900 °C and 30 mins just after the 500 °C and 4 mins?
- ‘After applying 500 °C and 2 min …. shown in Figure 25)’ (Page no14, line no 271 to 274). This is a very nice observation. However, it is just the experimental result only. What are the physical explanations for this result?
- Figures:
- There are too many figures. Some of these can be merged. For example, figure 8(a) and (b). Also, check the titles of the axes (figure 8(a)).
- Figure 9, the limit of the x-axis (2q) can be adjusted to see the difference between annealed and non-annealed curves.
- The text in Figure 20 is not clear.
- The descriptions of the sub-figures are missing in the text (most of the cases). For example, figure 21(a) and (b)
- S1, S2, and S3 are not well defined in the manuscript.
- Some errors: There are several errors. For example, i) check the legend of figure 7. There is only one curve.
- ii) The full form of HEMT (high electron mobility transistor) is missing.
iii) Page no 11, line no 223, ‘±20V’. Should be ‘±20 V’.
- References: Formats of the references are not the same. Please check the reference no 2, 5, 6, 13, and 18.
For these inaccuracies, I cannot give a positive recommendation to the publication even though the results are interesting.

Reviewer 4 Report
Reviewer-Report attached

Round 2
Reviewer 2 Report
Although the paper presents the results of experiments that in my opinion put novel findings to a small extent to the knowledge the paper can be eventually accepted for publication.
Reviewer 3 Report
The authors have successfully addressed my comments. The manuscript is recommended for publication.
Reviewer 4 Report
file attached
